# Inducible Pluripotent Stem Cells as a Potential Cure for Diabetes

**DOI:** 10.3390/cells10020278

**Published:** 2021-01-30

**Authors:** Kevin Verhoeff, Sarah J. Henschke, Braulio A. Marfil-Garza, Nidheesh Dadheech, Andrew Mark James Shapiro

**Affiliations:** 1Department of Surgery, University of Alberta, Edmonton, AB T6G 2B7, Canada; marfilga@ualberta.ca; 2Department of Emergency Medicine, University of Saskatchewan, Saskatoon, SK S7N 0W8, Canada; shenschk@ualberta.ca; 3Alberta Diabetes Institute, University of Alberta, Edmonton, AB T6G 2B7, Canada; dadheech@ualberta.ca; 4FRCS (Eng) FRCSC MSM FCAHS, Clinical Islet Transplant Program, Alberta Diabetes Institute, Department of Surgery, Canadian National Transplant Research Program, Edmonton, AB T6G 2B7, Canada; jshapiro@ualberta.ca

**Keywords:** islet cell transplant, diabetes, inducible pluripotent stem cells, immunosuppression, immune reset, insulin

## Abstract

Over the last century, diabetes has been treated with subcutaneous insulin, a discovery that enabled patients to forego death from hyperglycemia. Despite novel insulin formulations, patients with diabetes continue to suffer morbidity and mortality with unsustainable costs to the health care system. Continuous glucose monitoring, wearable insulin pumps, and closed-loop artificial pancreas systems represent an advance, but still fail to recreate physiologic euglycemia and are not universally available. Islet cell transplantation has evolved into a successful modality for treating a subset of patients with ‘brittle’ diabetes but is limited by organ donor supply and immunosuppression requirements. A novel approach involves generating autologous or immune-protected islet cells for transplant from inducible pluripotent stem cells to eliminate detrimental immune responses and organ supply limitations. In this review, we briefly discuss novel mechanisms for subcutaneous insulin delivery and define their shortfalls. We describe embryological development and physiology of islets to better understand their role in glycemic control and, finally, discuss cell-based therapies for diabetes and barriers to widespread use. In response to these barriers, we present the promise of stem cell therapy, and review the current gaps requiring solutions to enable widespread use of stem cells as a potential cure for diabetes.

## 1. Insulin as a Treatment, Not a Cure

In 1889, Oskar Minkowski and Joseph von Mering completed a canine pancreatectomy and induced fatal diabetes mellitus (DM). This experiment demonstrated the central role of the pancreas in glycemic control [1]. In 1893, Williams and Harsant working in Bristol, UK, attempted to transplant pancreatic fragments taken from a freshly slaughtered sheep and placed them subcutaneously in a boy dying of diabetic ketoacidosis, with unsuccessful results [2]. Even throughout the journey to discover insulin, Banting’s initial trials focused on subcutaneous injection of an unpurified pancreatic slurry, and the first patient treated developed a sterile buttock abscess [3]. Although Banting, Best, Collip and Macleod subsequently prepared more purified insulin extracts using acid-alcohol to dissolve the insulin and prevent degradation by exocrine enzymes, Banting’s acceptance speech for the 1923 Nobel Prize in Physiology and Medicine concluded with these words:

“Insulin is not a cure for diabetes; it is a treatment. It enables the diabetic to burn sufficient carbohydrates, so that proteins and fats may be added to the diet in sufficient quantities to provide energy for the economic burdens of life [3].”

Nearly 100 years later, this remains true. Despite novel, improved recombinant insulin formulations, the potential of “smart” insulins that are inactivated in a hypoglycemic environment, the advent of continuous glucose monitoring (CGM) and wearable biomechanical closed-loop pancreas systems, subcutaneous insulin remains a highly problematic treatment. The United States type 1 DM (DM1) exchange registry with >20,000 participants from 2016–2018 demonstrated that only 21% of adults and 17% of children achieve the recommended HbA1c goal of <7 and 7.5%, respectively [4,5]. Current HbA1c levels of 9.0% in 13–17-year-olds are only marginally lower with novel treatment options than the 9.5% seen in the same population during the 1980s [4,5]. Hypoglycemia also remains a significant but often overlooked complication of DM. Hypoglycemia occurs in 31–41% of diabetic patients [6], often at night due to the four-fold variability of overnight insulin requirements [7,8,9]. Of 11,061 exchange registry respondents, 6% reported hypoglycemic seizure or loss of consciousness within the previous three months-a risk that increases with age and the presence of hypoglycemic unawareness [4,10]. These events may be life threatening, with an incidence of 320 episodes per 100-patient years in patients that have lived with DM1 for more than 15 years [11]. Unfortunately, this risk escalates with intensive insulin therapy and improved control of hyperglycemia [11]. Achieving euglycemia is nearly impossible without flexible, dynamic insulin and glucagon responses and even the most advanced insulin therapies still fail to recreate the precise and physiologic glycemic control orchestrated by almost three million pancreatic islets of Langerhans.

This review briefly discusses novel insulin-based therapies but focuses primarily on the future promise of a potential cure for DM using cell-based therapies and islet stem cell transplantation (ISCT). We review novel mechanisms for insulin delivery and describe their shortfalls. We describe in vivo and in vitro islet cell embryological development and physiology to better understand its implications in the generation of functional stem cell-derived islet cells. Finally, we discuss the evolution of islet cell transplantation (ICT) as a cell-based cure for DM and its barriers to widespread use, as well as its importance in the future of stem cell-based therapies. Finally, we present a response to these barriers and review the current gaps requiring further research to enable widespread use of cell-based therapies, including pluripotent stem cells, as a cure for DM.

## 2. Novel Subcutaneous Insulin Delivery

The use of CGM, continuous subcutaneous insulin infusion (CSII i.e., insulin pump), and closed-loop wearable insulin delivery (i.e., artificial pancreas) devices has increased substantially in recent years, but are still only accessible to a relatively small subset of patients with DM. From 2011 to 2017, CGM use increased from 7 to 30% and CSII from 57 to 63% [4]. CGM, CSII, and artificial pancreas technologies all demonstrate lower HbA1c levels compared to standard insulin treatment [4]. CGM alone improves DM understanding and glycemic control. It guides novel treatment modalities and glycemic optimization by demonstrating real time glycemic targets and time spent in euglycemia, hypoglycemia, or hyperglycemia [5,12,13]. CGM also provides overnight and dynamic readings, and offers hypoglycemic and hyperglycemic alarms. Both independently, and combined with novel insulin delivery tools, CGM users have improved glycemic stability [4]. Advances in wearable insulin pump technologies have also shown clear benefits. A large meta-analysis of 33 randomized controlled trials demonstrated improved glycemic control with CSII compared to standard insulin delivery methods [14]. Bekiari et al. conducted a further meta-analysis comparing artificial pancreas to other forms of insulin therapies, including CSII, and showed the greatest glucose stability using dual hormone artificial pancreas devices [14,15]. Improved overnight glycemic control with artificial pancreas therapy was especially notable, as this has historically been difficult to manage with subcutaneous insulin [14,15]. For those who can access and afford these technologies (CGM, CSII, and closed-loop wearable insulin delivery devices), DM care is clearly improved.

However, despite enhanced glycemic control offered by CGM, CSII, and artificial pancreas technologies, they remain far from a cure (Table 1). HbA1c reductions with CSII, although statistically significant, are only 0.3–0.7% [4,14,15]. Even with fully automated, dual hormone artificial pancreas treatment, average daily and overnight glucose improved by only 0.48 mmol/L and 0.81mmol/L, respectively, compared to standard insulin therapy [15]. Additionally, normoglycemia was only achieved 16.4% of the time for patients using an artificial pancreas technology [15]. When provided structured DM training, patients can achieve similar glycemic control, decreased incidence of hypoglycemia, and improved psychosocial outcomes using self-directed subcutaneous insulin therapy compared to those with CSII [16]. Technical barriers also persist—issues with absorption, lipohypertrophy, rashes and skin reactions from the adhesive devices and extended use in one site can lead to progressively worse glycemic control despite automated insulin delivery [17,18,19]. Mechanical failure of infusion systems occurs frequently, with catheter kinking or occlusion, leaking, bruising, or infection at the site of insulin instillation occurring in up to 64% of devices over 7 days [5,19,20,21]. The biggest risk involves unrecognized discontinuity of insulin delivery, which occurs regardless of the needle/injection type, and may lead to diabetic ketoacidosis [20,21,22,23,24]. CGM and CSII also have patient-related factors limiting their utility. Even the most automated artificial pancreas systems require user input for bolus dosing and mechanical errors can occur due to patient misunderstanding or misuse [21]. Additionally, 47% of patients report device discomfort as a barrier to use, and 35% dislike devices on their body [5,25]. Others have reported skin irritation [26], and sleep disruption from bedtime alarms as problems [5,27]. While these therapies offer specific glycemic benefits, the absolute benefit, reliability, and usability concerns limit optimism (Table 1).

## 3. Islets of Langerhans

Current injectable insulin technologies fail to recreate physiologic glycemic control with a tight 1–2 mmol/L glycemic variance. In situ physiologic intraportal hormone delivery from the pancreatic islets of Langerhans maintains basal normoglycemia with insulin and counterbalances hypoglycemia with glucagon. Insulin output can increase up ten-fold after a meal, and return rapidly to basal levels with no hysteresis. In our opinion, exogenous subcutaneous insulin delivery, even when provided by the most ideal closed loop systems, cannot recreate this degree of dynamic control. Thus, developing a cell-based cure through islet cell generation and transplantation remains an ideal to strive for. Achieving this goal, especially with stem cell therapies, demands complete understanding of embryological differentiation and physiology of the islets of Langerhans.

### 3.1. Embryological Development and Structure

Islets form collections of cells that exist uniformly throughout the pancreas but represent only 1–4%, 2 g, or 2 mL of the pancreatic volume [28]. Person-to-person heterogeneity is common but islets are generally composed of approximately 60% β-cells, 30% α-cells, <10% δ-cells, <5% γ and ε cells producing insulin, glucagon, somatostatin, pancreatic polypeptide, and grehlin, respectively [28]. Islet mass varies throughout life, and expands during childhood growth and during normal pregnancy. The exact mechanisms that regulate this expansion process remain incompletely understood.

Mature β-cells develop from embryonic stem cells (hESC) in a continual process that may be considered in seven steps beginning from definitive endoderm, to primitive gut tube, posterior foregut, pancreatic endoderm, endocrine precursors, immature β-cells, and finally mature β-cells (Figure 1) [29,30,31]. Definitive endoderm forms during gastrulation from epiblast cells undergoing epithelial to mesenchymal transition [32]. This process is initiated by Wnt3a protein signaling [29,31,33,34], followed by Nodal signaling-mediated activation of the TGFβ pathway that ultimately leads to activation of intracellular Smad2 and differentiation into the primitive streak and definitive endoderm [35,36,37]. Stable, bioactive Nodal does not exist. Fortunately, a similar protein from the TGFβ family, activin-A, acts as an in vitro biochemical analogue to activate Smad2 [38,39,40]. In vitro, hESC exposure to Wnt3a and activin-A leads to 95% definitive endoderm cells that express the phenotypic markers SOX17 and FOXA2 [41]. Patterning of anterior–posterior axis occurs with exposure to KGF/FGF7 and creates the primitive gut tube [29,30,34]. Subsequent culture with B27 supplement, retinoic acid, Noggin, and a smoothened (Smo) inhibitor, such as cyclopamine or Sant 1–4 molecules to prevent Hedgehog (Hh) signaling, induces differentiation into the posterior foregut that has potential to become pancreatic, hepatic, or duodenal tissues [29,34,42]. Hepatic tissues are favored through bone morphogenetic protein (BMP) signaling pathways, while endocrine differentiation is blocked by FGF10 activation [43,44]. Exposure to Noggin or LDN193189, potent inhibitors of both BMP and FGF10, produces pancreatic endoderm cells (PDX1^+^) [29,34,43].

Further differentiation of pancreatic endoderm cells into islets has been incompletely understood and until recently, only occurred in three-dimensional (3D) culture in vitro [31,41,45]. Differentiation into pancreatic endocrine progenitors (PDX1^+^/NKX6.1^+^) utilizes TGFβ receptor I (TβRI/ALK5) inhibition and continued prevention of Hh signaling with Sant1–4 moecules [30,31,41]. Recent data has helped clarify why 3D culture and in vivo differentiation is required at this stage for β-cell differentiation. Failure to produce NKX6.1^+^ cells prior to expression of endocrine genes such as neurogenin 3 (NEUROG3) produces non-functional poly-hormonal cells [30,46]. Hogrebe et al. (2020) recently demonstrated that the cellular microenvironment, actin cytoskeleton, and cellular attachments, dictate NEUROG3 expression [30]. Firm adhesion of stage 4 (PDX1^+^) cells to Type-I collagen coated culture plates leads to NKX6.1^+^ cells, followed by stage 5 actin depolymerization with latrunculin A to allow NEUROG3 expression. Similarly, inhibition of YAP1 function increases NEUROG3 expression and favors endocrinogenesis [47]. Further maturation leads to insulin producing, NKX6.1 expressing, β-cells with islet-like glucose response in vivo [30].

Maturation and differentiation specificity and efficiency may be further improved with various compounds (Figure 1). It should be noted that use and timing of these compounds varies widely by protocol. CHIR99021, a selective glycogen synthase kinase-3β inhibitor, has been used in stage 1 formation of definitive endoderm to increase cell viability [30,31,48]. Rezania et al. added vitamin C from the primitive gut tube to pancreatic endoderm (stages 2–4), to increase cell numbers and confluency and reduce NGN3 expression, which has demonstrated disruption of pancreatic endoderm [31,49]. Increased protein kinase C activity, in vitro with -(*2S,5S*)-(*E,E*)-8-(5-(4-(trifluoromethyl)phenyl)-2,4-pentadienoylamino)benzolactam (TPPB) has demonstrated improved induction of pancreatic progenitors from primitive gut tube (stages 3–4) [29,30,50,51,52]. Thyroid hormone acts after stage 5 through the transcription factor MAFA to improve glucose-responsive insulin release in mature cells [30,31,41,53]. Alongside thyroid hormone, gamma secretase XX inhibitor (XXi), which inhibits the Notch pathway and increases NGN3 expression, has been used in step 6 to inhibit PTF1a guided exocrine differentiation to improve β-cell maturation [30,31,38,41,43,49]. Application of these compounds is not standardized, and no author to date has combined all these additives to determine if an ideal, more efficient or specific β-cell differentiation can be achieved. Greater understanding of their role, result replication, and process standardization are needed to determine ideal additive compounds.

A major limitation in our understanding of islet developmental science is that many of the concepts and protocols have been derived from work in murine models—mainly because the relevant human targets and growth factors have yet to be defined. While there may be conservation in the early developmental pathways between species, it seems unlikely that this process will be fully optimized until these pathways are mapped out entirely in human cells. Another important limitation of in vitro islet generation is that it only approximates but does not replicate the continuum of cell-to-cell contact, dynamic intracellular signaling and participation of the physiologic extracellular matrix present in the full complexity of a developing human embryo. Only when we can recapitulate the process with more accuracy will we be able to optimize, perfect and avoid risk of off-target cell growth in this differentiation process.

### 3.2. Function

Glucose control is accomplished with both autonomic nervous and hormonal systems (Figure 2). While interest focuses on β-cells, the α, δ, γ, and ε cells also play increasingly well understood and important roles in glycemic control.

In the fasting state, normoglycemia is achieved through activation of the autonomic nervous system; sympathetic activation leads to glucagon release from α-cells, while parasympathetic activity induces insulin release from β-cells [54]. These actions are directed through glucose-sensing cells located in peripheral locations such as the hepatoportal vein area, and by specialized glucose-excited or glucose-inhibited neurons located in the hypothalamus or brainstem region [54]. This mechanism directs α and β-cells to release basal levels of glucagon and insulin to promote appropriate hepatic gluconeogenesis for anabolism and cellular functions [54]. In anticipation of food, either by sight, mastication, or gastric distention, and prior to any blood glucose changes, parasympathetic release of acetylcholine activates β-cell muscarinic receptors (m3AchR), producing phospholipid-derived messengers to initiate protein kinase C (PKC) directed calcium influx and efficient insulin release through the cephalic response [54,55].

Elevated blood glucose concentrations lead to biphasic insulin release, lasting approximately 60 min [5,56]. The first phase occurs with GLUT2 facilitated diffusion of glucose into β-cells, which is oxidatively metabolized to produce ATP. In response, ATP dependent K^+^ channels (KATP) channels close, leading to cellular membrane depolarization and opening of voltage-dependent L-type calcium channels. Intracellular calcium promotes SNARE protein mediated exocytosis of insulin-containing secretory granules with release into portal circulation (Figure 2) [5,56]. Depolarization and exocytosis oscillate every 3–6 min to avoid insulin receptor downregulation [5,56].

A second phase of insulin release, accounting for approximately 50% of postprandial insulin secretion, occurs via stimulation from parasympathetic inputs, glucagon-like peptide 1 (GLP-1), glucose-dependent insulinotropic peptide (GIP), free fatty acids (FFA), and somatostatin (Figure 2) [5,56,57,58,59]. GLP-1 and GIP are incretins secreted from pancreatic α-cells, as well as K-cells and L-cells located in the pancreas, ileum, and colonic bowel in response to increase blood glucose concentration [56,59]. This demonstrates the α-cell interaction with β-cells to achieve euglycemia. GLP-1 and GIP act through β-cell G-protein-coupled receptors (GPCR), increasing 3′,5′-cyclic adenosine monophosphate (cAMP) and leading to protein kinase A (PKA) dependent and non-PKA dependent insulin exocytosis [57,59]. Similarly, FFA act through the GP40 GPCR to further stimulate insulin release [57,58]. δ-cell released somatostatin, and γ-cell released pancreatic polypeptide, also play a minor role for glucose homeostasis but mechanisms for such are incompletely understood [60,61]. Ablation of δ-cells impairs islet cell function [60], and infusion of pancreatic polypeptide alongside insulin reduces insulin requirements [62]—further analysis of these mechanisms may assist with improving glycemic control but also highlight the complex interplay of cells required for glycemic control that is often overlooked with single or dual hormone treatment systems.

## 4. Islet Cell Transplantation

In 2000, Shapiro et al. revolutionized clinical outcomes with ICT demonstrating proof-of-concept that cell-based therapy could offer huge potential for the treatment of DM. Their results demonstrated 100% insulin independence at one year in seven DM1 patients consecutively treated with glucocorticoid-free immunosuppression using anti-CD25 monoclonal antibody (mAb) induction immunosuppression and maintenance immunosuppression with tacrolimus and sirolimus [63,64]. Unfortunately, long-term insulin independence was not achieved, with most patients returned to low doses of insulin over time. Protocol improvements now demonstrate five-year ICT insulin independence rates >50%, matching rates observed with whole pancreas transplant, but with significant less morbidity after ICT [11,65,66]. Within-subject, paired comparison of insulin injection versus CSII, and CSII versus ICT demonstrated stepwise improvement of glycemic control, less glycemic variability, and fewer hypoglycemic events, with the best results achieved after ICT [67]. Notably, HbA1c improved from 8.2% using CSII to 6.4% with ICT [67]. Glycemic stability and a lower incidence of hypoglycemia also persisted following ICT regardless of insulin independence [67]. Multicenter phase III clinical trial data also demonstrated that 87.5% and 71% of patients, at one and two years’ post-transplant, respectively, achieved a HbA1c <7.0% and median HbA1c of 5.6% [68]. Similar HbA1c results were observed by the Vancouver group with HbA1c of 6.6% following ICT versus 7.5% with intensive insulin treatment; they also reported significantly less retinopathy, nephropathy, and a trend towards less neuropathy with ICT [69,70]. Others have also demonstrated improved retinal blood flow and improved markers of polyneuropathy after ICT [71,72].

ICT has revolutionized the care of patients with DM, with benefits beyond hyperglycemic control. These results have been achieved through optimizing multi donor transplantation, islet isolation [66], good manufacturing practices (GMP) [66,73], and agents to resist the immune and non-immune challenges presented in Figure 3. Detailed GMP-islet isolation procedures have been made available from the clinical islet transplantation consortium, allowing clinical isolation facilities to utilize >50% of donated organs [66,74]. Once isolated, current islet cell culture techniques have allowed a substantial decrease in the number of apoptotic cells and minimized harmful cytokine release following transplant [75,76,77,78]. Specific agents to mitigate inflammation and apoptosis have also increased ICT clinical success, including the interleukin 1 antagonist anakinra and TNF-α inhibitor etanercept [65,66,79,80,81,82]. Similarly, adding manganese superoxide dismutase decreases reactive oxygen species and has shown to enhance in vitro islet cell viability with augments in vivo murine marginal islet mass engraftment [83,84]. An improved understanding of the blood-mediated inflammatory reaction (IBMIR) following ICT has led to post-ICT heparin infusion to limit tissue factor-related IBMIR, while insulin infusion allows islet rest and reduced inflammation to improve engraftment [85,86]. Finally, depleting T-cell populations with induction therapy using alemtuzumab or thymoglobulin has been more effective that IL-2 receptor (anti-CD25) blockade with less potent daclizumab or basiliximab. All these additions have contributed to enhanced long-term insulin independence rates [66,79]. Other agents that may further improve ICT engraftment and success include liraglutide or pan-caspase inhibitors to further improve insulin independence rates [11,65,87,88,89,90]. Ongoing research promises to elucidate additional modifications to improve graft success. Immunogenic protection with regulatory T cells (Tregs) may enable optimal engraftment and a decrease (or complete elimination) of lifelong pharmacologic immunosuppression [91,92]. Achieving this would closely resemble a true cure for DM.

### Barriers to Islet Cell Transplant

Despite excitement, numerous barriers to widespread use of ICT use persist. The only current islet cell source is human deceased donor pancreata, and the supply of potential organ donors is severely limited in the context of the prevalence of DM. Each recipient requires >5000 islet equivalents (IEQ) per kg and ideally >11,000 IEQ/kg for insulin independence, and typically 2–4 pancreata per recipient, further straining a small donor pool [66]. Access is also limited by funding. In 2012, the only countries that funded ICT under non-research, clinical care streams were Canada, Australia, the United Kingdom, France, Switzerland, Norway, Sweden, and parts of Europe [11,93]. Even in countries with access, lifelong immunosuppression requirements and associated complications mean that strict recipient criteria must be met for islet-alone transplant (i.e., without kidney). Patients must have recurrent severe hypoglycemic episodes with hypoglycemic unawareness, glycemic lability not managed with intensive insulin, pumps and/or continuous glucose monitoring therapies [11]. They should also have had DM1 for >5 years, be over the age of 18, have normal renal function, and have a BMI (<30 kg/m^2^) and/or weight <90 kg and/or daily insulin requirement <1.0 U/kg.

Even when patients access ICT, alloimmunity, and autoimmunity in type 1 diabetes (DM1), mean patients must remain on lifelong potent immunosuppression. Infectious risks and toxic effects from immunosuppression have improved but persist and must be balanced against ICT benefits. Timing of the ICT must also be considered, as earlier ICT prior to diabetic complications is ideal but increases length of immunosuppression exposure, in-turn increasing the risk of infection, cancer and drug toxicity. Risk of opportunistic infections include cytomegalovirus (15%), cytomegalovirus retinitis (20%), varicella zoster (5%), and nocardia (2%), amongst other infections [94,95]. Severe infection remains rare and more commonly patients experience minor concerns including acne, mouth ulcers, and diarrhea [94]. Calcineurin-inhibitors are especially notable in that they are both nephrotoxic and diabetogenic [65,96,97]. Malignancy, namely squamous and basal cell carcinoma of the skin, occurs in 2% of ICT patients, and post-transplant lymphoproliferative disorder (PTLD) occurs in approximately 1%. Mortality related to immunosuppression in the context of ICT is 0.19% [66,98]. These risks occur despite approximately 50% of ICT failing to achieve long-term insulin independence. Insulin independence is limited by auto- and alloimmunity, but also imperfect engraftment that decreases functional islet cell mass. Engraftment is limited by apoptosis, thrombosis, ischemia, inflammation, and instant blood-mediated inflammatory reaction (IBMIR) [66,99,100]. While many ICT recipients benefit irrespective of complete insulin independence, dynamic risk-benefit analysis should be contemplated and individualized in every case. Considering evolution of artificial pancreas technology, carefully designed randomized control trials with intention-to-treat analysis are required to compare ICT to novel subcutaneous delivery systems.

## 5. The Promise and Future Challenges for Stem Cells

Islet/β-cell stem cell-derived therapies offer the potential to overcome many of the barriers emphasized above to widespread application of ICT. ISCT has the potential to resolve limited access, donor shortage, and need for immunosuppression. Human embryonic stem cells (hESC) and inducible pluripotent stem cells (iPSC) can be differentiated into mature β-cells that co-express PDX1, NKX6.1, MAFA, Insulin, C-peptide, that have prohormone processing enzymes, and most importantly, that are glucose responsive in vivo [29,30,31,34,41]. Stem cell differentiation and expansion can now occur in 2D and 3D growth media following the seven-step embryological process shown in Figure 1, with resultant islet-like cell clusters capable of consistently reversing diabetes in murine models [30,31,41].

Specific challenges relating to hESC/iPSC-based ISCT approaches remain if this therapy is to one day be applied as a widespread cure for all forms of DM. Determining the ideal source for islet generating stem cells (allogeneic versus autologous iPSC), the optimal transplant site, and identifying an approach to eliminate immunoreactivity remain unanswered. Lastly, if these therapies are to be used as a true cure for DM, economically viable scale up and supply with standardized GMP protocols to generate these cells is vital (Figure 4).

### 5.1. Stem Cell Source

The two primary sources of iPSC are allogeneic and autologous, both offer benefits and drawbacks. Allogeneic sources allow for mass generation of islet-like cells from a single, optimized iPSC source. Large pools of HLA-specific iPSC-generated cell lines could be generated to provide ‘haploidentical’ islet-like cells. This may offer a homogenous cell source with optimal glycemic control, less off-target growth, and easily accessible HLA-matched islets for ISCT. However, despite major HLA matching, patients will certainly require some degree of immunosuppression as inevitable minor HLA mismatches will still generate immunoreactivity if not otherwise modified [101]. The most significant barrier to allogeneic transplant is therefore immunoreactivity and post-ISCT immunosuppression requirements. ViaCyte (previously NovoCell) has been at the forefront of technologies attempting to resolve this barrier. They hope to demonstrate successful allogeneic ISCT engraftment, hormonal release, and immunoprotection through clinical trials evaluating the PEC-Encap (VC01) and PEC-Direct (VC02) devices. Albeit these clinical trials use hESCs as the cellular substrate for differentiation and transplant, outcomes obtained from these groundbreaking efforts could prove valuable for iPSC-based therapies. The VC01 is a planar subcutaneous macro-encapsulation device for pancreatic progenitors with oxygen and nutrient transport capacity but also allo- and auto-immunoprotection to enable ISCT without immunosuppression [102]. Phase 1/2 clinical trials have demonstrated pancreatic progenitor maturation without off-target growth. In ViaCyte’s most recent clinical trials summarized in oral form, up to one third of patients demonstrated detectable human C-peptide in peripheral blood in previously C-peptide negative individuals with DM1. This correlated strongly with the persistence of polyhormonal insulin-expressing islet cells contained within the subcutaneous devices over time (unpublished data). The perforated VC02 device does not provide immunoprotection, but ongoing clinical trials are assessing efficacy of pancreatic progenitors to provide in-human insulin independence (Clinicaltrials.gov Identifier: NCT03163511). Although long-term results were limited by the foreign body response [29,103,104,105,106]; discovery of novel biomaterials for encapsulation that abrogate this reaction would provide promise for immunosuppression-free ISCT. Anderson et al. (2020) have demonstrated long-term insulin release in immunocompetent mice, without immunosuppression requirements or foreign body response, using microspheres and selectively permeable silicone devices coated with a synthetic polymer [107,108]. Previously successful microsphere and synthetic polymers that have enabled islet cell survival and immunoprotection in murine models have failed in humans due to a vigorous foreign body response [109]. Testing novel polymers in humans will certainly be required [107].

Autologous iPSC islet cell generation may allow for personalized ISCT. Islet cell auto-transplantation following pancreatectomy in the context of chronic pancreatitis is a crude first representation of the potential of this approach. Zhao et al. have raised concerns that islet cell maturation may alter cellular immunogenicity and thereby confer acute rejection [110]. Further investigation has revealed that immunoreactivity is only conferred in retrovirus-derived iPSCs due to leakage of transgenes and activation of neighboring genes, whereas plasmid derived iPSCs demonstrate negligible immune reaction [111,112,113]. iPSC-based ISCT without immunosuppression requirements would, therefore, be technically possible but remains to be tested. The costs and time of generating person-specific iPSC and then maturing them into islet-like cell clusters confers an astronomical barrier, but the hope is that with economies of scale, process automation and increased efficiency, mass iPSC-based ISCT manufacture will indeed be possible at reasonable cost. This will be critically dependent on advances in robotic engineering, artificial intelligence and machine learning, and collaboration with industry to take this from single patient to mass manufacture over time. HLA screening of individual autologous iPSC-based islet cell clusters would be another barrier. Rather than a single screened HLA-specific pool of transplantable iPSC-based islet cells, each patients autologous iPSC and matured islets would requiring screening for genetic mutations and off target growth prior to transplantation [113]. Additionally, variability exists between different iPSC lines, mostly due to genetic background differences, and their ability to differentiate into functional cells of a given lineage [114,115]. Overall, a better understanding of the in vivo immune response to HLA-matched iPSC islet cell clusters, or other alternatives to immune acceptance (as discussed below), and calculation of the cost/time feasibility and optimization for personalized autologous ISCT is needed to better determine the best source of iPSC-based islet cells.

Xenogeneic islet cell sources should not be overlooked. We have not reviewed them thoroughly here, but O’Connell et al. (2013) provide a complete review of this solution to ICT [116]. It is important to be aware that xenogeneic (porcine) sources provide a potentially large source of mature, insulin producing islet cells for transplantation. Two concerns for this islet cell source are xenogeneic immune reaction and the risk of zoonotic infection of porcine endogenous retrovirus. Genetic engineering and encapsulation devices have been utilized to prevent these reactions and clinical trials may be within reach [117,118,119,120,121]. Two trials by a single group have evaluated encapsulated porcine ICT, both showing potential therapeutic benefit; unfortunately, this is yet to be replicated by others [122,123]. Despite advances, xenogeneic sources currently remain futuristic and require replication and larger scale studies to evaluate their clinical benefit.

### 5.2. Transplant Sites

The ideal implantation site for ISCT should first and foremost provide hormone release in a physiologic location; other desired characteristics in decreasing importance include vascular and environmental support for islet cell engraftment, easy access for transplantation, immunoprotection, and retrieval capacity. Potential sites include renal subcapsular, subcutaneous (within devices or modified spaces), omental, and intraportal [124,125]. The renal subcapsular space has demonstrated promising results in murine models, but has failed to achieve euglycemia due to limited subcapsular space and exocrine contamination in larger animals and humans [125,126,127]. The subcutaneous and intramuscular sites have been also investigated due to their easy transplant procedures, easy resection in case of off-target growth, and easy monitoring with non-invasive imaging. Major issues include non-physiologic release of hormones, poor vascular and environmental islet cell support, and immunoprotection. Recent unpublished results from ViaCyte are encouraging and the PEC-Encap (VC01) and PEC-Direct (VC02) devices may enable viable subcutaneous ISCT, as long as the foreign body response can be mitigated through the use of novel biomaterials. Alternatively, Pepper et al. (2017) utilized this foreign body response to create a subcutaneous transplant site with neovascularization and collagen support for islet cell engraftment [124]. This technique enables optimized subcutaneous engraftment; however, immunosuppression remains a barrier to its applicability for widespread use. Overall, encapsulation devices offer a unique tool to study iPSC islet cell maturation and insulin release for DM reversal. Their greatest benefit is enabling in-human evaluation of off-target growth with easily retrievable devices and demonstrating applicability of iPSC islet cell cluster maturation and survival in vivo.

Insulin independence necessitates adequate islet engraftment without fibrosis, which is currently only offered with omentum and intraportal ICT. Omental ICT has demonstrated positive early results in animal and human studies [125,128,129]. Stice et al. (2018) demonstrated successful omental autologous ICT in four patients confirming prior promising animal studies. The omentum releases hormones into portal circulation, supports islet engraftment, and is relatively accessible if resection is needed [128,129]. The omentum also limits IBMIR, since no direct blood contact occurs [129]. A limitation of this site is that surgical placement is required, which may limit widespread use due to cost and access to operative time, but all clinical trials to date involving omental implantation have used minimally invasive laparoscopic approaches [128,129]. Clinical trials have begun to further evaluate the omentum, but direct comparisons with the intraportal site are needed to guide future endeavors. Intraportal ICT remains the clinical gold standard because it has demonstrated adequate hormone release into the portal circulation, islet cell engraftment, and accessibility via radiologically-guided injection. Initial concerns regarding an 11% risk of portal venous thrombosis and bleeding following intraportal ICT has been diminished and nearly eliminated through well-described techniques that ablate the hepatic catheter tract and post-transplant heparinization to limit thrombosis [130,131]. The remaining barrier to intraportal iPSC-based ISCT, particularly due to the intrinsic inability to remove the infused islets from the liver, is uncertain off-target growth and teratoma formation. Off-target growth and teratoma formation has been demonstrated in 15–45% of cases when pancreatic progenitor cells were transplanted [29,34]. Off-target growth is likely reduced with more mature stage 6 cell-derived products, but longer term follow up is ongoing, as is investigation of treatment for off-target intraportal growth with ablation techniques [30,31]. Overall, the omentum and liver remain potential sites for transplantation, but larger in-human trials of omental transplant have yet to be completed; intraportal transplant remains the most viable long-term option with physiologic hormone release, adequate islet cell engraftment, easy transplantation techniques, and is only limited by the IBMIR, post-injection complications, and graft irretrievability. Novel genetic techniques to biochemically eliminate transplanted cells with kill switches may enable intraportal transplant without concerns for off target growth as we discuss below [132,133].

### 5.3. Immunoreactivity

Immunosuppression requirements remain one of the largest limitations to ICT. Autologous iPSC transplant offers a solution but may be limited due to its high costs. Alternatively, HLA-matched allogeneic iPSC-based ISCT would still require immunosuppression–likely at least as potent as current immunosuppression protocols used in islet transplantation today [100]. Approaches to managing or eliminating immunoreactivity for allogeneic iPSC islet cells are under examination. Liu et al. demonstrated that sourcing iPSCs from less immunogenic sources, such as umbilical mesenchymal cells instead of skin fibroblasts, could limit immunogenicity [134]. These iPSCs had statistically significant less immune reactivity, with less HLA expression and less T-cell expression of perforin and granzyme B, but results were modest and unlikely to enable immunosuppression-free transplant [134]. Micro and macro encapsulation allow immunoprotection for first-in-human safety and off target growth assessment, but are unlikely to provide a long-term solution with metabolic control and insulin independence due to the foreign body response. More definitive options for eliminating immunosuppression include immunomodulation, and iPSC gene editing.

Gene editing may offer the most robust option for eliminating immunosuppression requirements. It benefits from leaving the recipient’s immune system untouched and capable of immune regulation and effective infection control. The CRISPR/Cas9 system has been widely used to create and study genetic disease states such as Rett syndrome [135], HIV [136], and Parkinson’s [137], but has also been used to modify iPSCs and reverse genetic disease states in vitro [138,139,140]. These techniques may allow transplanted islet cell expression of tolerogenic cytokines, and immunomodulatory proteins. Increased interleukin-10 (IL-10) expression has demonstrated less immune activation, and improved graft survival without immunosuppression, in animal models for liver, lung, and corneal autologous transplant [141,142,143]. However, results demonstrate that although graft rejection is limited, it still occurs. On the other hand, complete elimination of HLA class-I molecules from stem cells offers a cellular transplant source readily available to all recipients independent of their genetic background or HLA type. HLA-silenced iPSC lines have been generated by targeted disruption of both alleles of the Beta-2 Microglobulin gene, and produce non-reactive iPSC cells in lymphocyte reaction assays with retained ability to differentiate into multiple cell lineages [144,145,146]. Further analysis with HLA-silenced iPSC-based islet cell for transplantation is required to determine long-term efficacy.

Immunomodulation or immune protection with genetic alteration may also offer protection from autoimmune re-activation. In patients transplanted with autologous or allogeneic HLA-silenced islet cells, patients with DM1 will likely still suffer from autoimmune graft destruction. Exogenous IL-10 supplementation [147], and more recently, gene transfer and increased islet cell IL-10 expression has demonstrated delayed recurrence of DM after syngeneic islet transplantation [148,149,150]. Similarly, increased PD-L1 expression may block effector T-cell mediated islet destruction and prevent autoimmune re-activation after ISCT [151,152,153,154]. Both IL-10 and PD-L1 mechanisms typically occur in vivo through the action of Tregs [149,150]. Therefore, increasing this cell population could provide ISCTs a similar immune protection. Studies have demonstrated alloantigen-specific immunosuppressive capacity of Tregs after transplant [155], and clear GMP protocols now exist to generate protective Tregs specific for recipient alloantigen’s under GMP conditions [156]. Unfortunately, these protocols would require patients to receive numerous Treg doses to maintain autoimmune protection.

An alternative to exogenous Treg infusions is a technique termed “immune reset” where the inappropriately activated immune system is eliminated and replaced with one with decreased effector T cells and proportionally more Treg cells to eliminate islet cell autoimmunity. This method was first discovered through evaluation of bone marrow-derived hematopoietic and mesenchymal stem cells (BMSC) as a source of inducible islet cells [157]. Although iPSCs have largely supplanted BMSC as a stem cell source, evaluating the pathway of islet cell regeneration though BMSC transplant inadvertently led to immune reset discovery. Following experimentally induced DM in streptozotocin-treated mice [158,159], streptozotocin-treated rats [160], E2f1/E2f2 mutant mice [161], and non-insulin-dependent KKAy mice [162], early BMSC treatment induced DM reversal. Despite insulin production and DM reversal, Hasegawa et al. (2007) demonstrated that BMSC did not differentiate into islets but instead initiated islet regeneration from pre-existing pancreatic progenitor cells [163]. Voltarelli et al. (2007) tested these techniques clinically; they mobilized patient’s CD34^+^ (hematopoietic BMSC), collected them via leukopharesis, and then intensively immunosuppressed patients for five days with cyclophosphamide and rabbit antithymocyte globulin for immune ablation. CD34^+^ cells were then re-introduced to patients and 87% medication independence and 96% insulin-independence was achieved in 23 patients with newly diagnosed DM1 [157,164]. Evaluation of this technique demonstrated that it not only leads to maturation of pancreatic progenitor cells into islets, but also resets the immune system to prevent cytotoxic T-cell activation through extended duration CD4^+^ T cell depletion [165].

Current immune reset techniques do not offer long-term insulin independence, primarily due to recurrence of autoimmunity. However, we currently have an ongoing clinical trial in Edmonton that is exploring the potential of the drug plerixafor to mobilize CD34+ stem cells into the peripheral blood. This trial, approved for adults and adolescent children with new onset DM1 uses a single dose of T cell-depleting therapy, dual anti-inflammatory medications and a long-acting GLP-1 analogue to promote immune reset. Using this technique, BMSC are mobilized from a patient’s own bone marrow and may enable yearly doses to maintain autoimmune protection.

Genetic modification may also resolve other barriers to iPSC-based ISCT. Enabling non-immunogenic islet cells eliminates cost of personalized medicine but may also eliminate concerns regarding off-target growth. Off-target growth could be controlled using gene-edited cell lines with drug-inducible kill switches. Liang et al. (2018) demonstrated effective drug activation of an essential cell-division gene (CDK1), while Di Stasi et al. (2011) took this further and genetically expressed a drug-inducible caspase-9 (iCasp9) that allowed complete apoptosis of transplanted T-cells, even when they were not proliferating [132,133]. Incorporating a similar, inducible mechanism for apoptosis in iPSC islet cells has not been demonstrated, but would allow for mitigation of concerns for off-target growth and enable safe intraportal transplantation.

Many of these solutions to immunoreactivity have been proven but have yet to be trialed specifically for ISCT and work remains to be done. Combining allogeneic protection with HLA-silenced iPSCs, autoimmune protection with IL-10 or PD-L1 expression for Treg activity upregulation, and immune reset together provides promise immunosuppression free ISCT. Meanwhile, successfully demonstrating drug-induced apoptosis and safe intraportal transplantation may eliminate fears of off-target ISCT growth. This would allow for a single source of allogeneic, but HLA-silenced and autoimmune protected islet cells, with controlled safety switches to enable immunosuppression-free intraportal transplant.

### 5.4. Scale out, Scale up, and Increased Culture Surface per Volume

As we move closer to a cell-based cure for DM, a significant challenge will be providing them to >4 million DM1 patients [5,166]. A parallel can be drawn to chimeric antigen receptor (CAR)-T-cell oncologic immunotherapy; once CD19-targeted CAR-T-cell therapy demonstrated remarkable benefit for acute lymphoblastic leukemia, a significant bottleneck for widespread use developed [167,168]. Personalized CAR-T-cellular therapy demonstrates remarkable similarity to allogeneic iPSC-based therapies, with cell collection from patients, genetic modification using CAR cDNA and then subsequent cellular expansion and selection with quality control prior to patient use [167,168]. We expect a similar supply and demand bottleneck that will limit initial widespread use of iPSC-based ISCT once therapeutic benefit is demonstrated and the complex manufacture processes have been stabilized. This bottleneck will be amplified if iPSC sources are autologous, since each patient will require unique iPSC generation and expansion; however, even if allogeneic sources are used, few labs currently exist that can make GMP iPSCs-based islet cell clusters. A step-by-step approach to enable scale up and treatment of the hundreds of million patients with DM is needed. Learning from barriers faced by CAR T-cell therapy, we suspect that iPSC supply shortage may be overcome by generating a consistent GMP protocol for islet stem cell production, regionalization of iPSC-based islet cell generation, and technological solutions for mass production to create an economy of scale and inexpensive DM cure (Figure 5).

The first step of scale up will be consistently demonstrating a safe, and efficient GMP protocol for iPSC-based ISCT. CAR T-cell therapies initially struggled to achieve widespread use due to product heterogeneity caused by variability in “manufacturing processes, source materials, viral vectors, ancillary reagents, quality control, post-treatment immune monitoring, and government regulation [168].” iPSC-based ISCT technologies should use this experience as a learning opportunity to standardize GMP protocols including processes, reagents, and quality control. Current iPSC islet cell production is variable, especially with regards to additives to improve specificity and efficiency [30,31,41]. Having standardized processes will enable creation of consistent, homogenous products and facilitate government approval with common international production standards and regulations. Standardization will also enable definition of critical quality standards for and quality by design, which sets out required attributes of the final products and guarantee them via assurance of the design process rather than necessitating testing of each product, thus saving money [169]. Well defined, standardized GMP protocols will enable economically viable and consistent products for use.

Once standardized and approved, regionalization of processes should be then implemented. CAR T-cell therapy outcomes are limited by “vein to vein” time, whereas islets are capable of being preserved in culture [170]. CAR T-cell limitations initially forced patients to travel long distances for treatment, which reduced production capacity and made CAR T-cell transport difficult with specialized couriers required to maintain handling quality and proper therapy identification [170]. With standardized GMP protocols, iPSC generation and purification expertise may shift to regional centers to reduce laboratory production costs and ensure product consistency—a concept already proven in islet isolation for ICT [66,171,172,173,174]. This will require significant collaboration between the iPSC laboratories, transplant coordinators, researchers, technicians, physicians, and patients at the recipient’s center, but will significantly reduce costs of producing GMP stem cell-derived islet cell clusters [174].

Lastly, aligning production and remuneration with demand and healthcare budgets may be the final barrier to making ISCTs a first line therapy for DM. This will require creating economies of scale. Centralized production lowers costs by spreading the initial monetary investment of an approved GMP facility but limits production capacity. Maximizing the production capacity of centralized facilities will become of utmost importance. Doses for most cell-based therapies are approximately 10^7^ to 10^9^ cells [169,174]. As above, ICT requires minimum >5000 IEQ/kg and ideally >11,000 IEQ/kg for insulin independence [66]. Improved islet purity from the current 30–50% up to 100% with iPSC based islets will enable safe ISCT of minimum 13,000 IEQ/kg and potentially up to 20,000 IEQ/kg to allow for functional reserve and improved long-term insulin independence. With a standard 70 kg patient, and estimated 1000 cells/islet [175], at least 9.1 × 10^8^ to 1.4 × 10^9^ cells would be required. Engineering collaborations will be required to enable single laboratories to create these large volume cell therapies within the size constraints of a lab; this will be emphasized if autologous cells are used, where one person’s cells are reverted into iPSCs and then expanded exponentially prior to islet differentiation to provide a personalized cure, as opposed to a set number of HLA-matched, or HLA-silenced, allogeneic iPSC lines that could be expanded and banked. Keys to achieving an economy of scale with high-throughput and large-scale iPSC expansion (autologous or allogeneic) will be to identify an appropriate growth medium, extracellular matrix (ECM), and environment for mass production [176].

#### 5.4.1. Growth Medium

Growth media provides important nutrients and cell signaling factors for iPSC expansion and differentiation. An ideal growth medium for commercialization would allow cheap, ethical, and easily reproducible products to be formed. This largely eliminates serum and animal-sourced media. Historically, fetal bovine serum was required for stem cell expansion but more recently, Chen et al. (2011) described the TeSR-E8 medium, which allows growth of various iPSC lines with improved reprogramming and experimental consistency [177]. Sui et al. (2018) have used a similar media (StemFlex), which claims to have fewer components and enables superior single-cell passaging, gene editing, and reprogramming [41]. Direct comparison of these two media is required to help guide iPSC expansion standardization.

#### 5.4.2. Extracellular Matrix

Until recently, 2D culture was the only method for stem cell expansion. Stem cells grow as adherent colonies and upon detachment, degrade into embryoid bodies (EB) [178]. This is due to the requirement of ECM-integrin interactions to maintain pluripotency and continued expansion. Matrigel and Geltrex are two commonly used basement matrices used to allow ECM interactions in cell culture for iPSC expansion [30,31,41]. Unfortunately, these are semi-chemically defined, xenogeneic substrates, that are difficult to sterilize with standard techniques and have significant variability limiting them from clinical use [176,178]. Growth with recombinant laminin-511, a xeno-free recombinant protein, represented a significant advancement of our understanding of ECM importance [176,179]. It led to the discovery that the interaction between laminin -111, -332, and -511 and its primary receptor integrin α6β1, supports stem cell expansion and blocks differentiation into EB [178,180]. Unfortunately, many of these recombinant protein surfaces were limited by cost, therefore, novel synthetic surfaces have been developed to mimic these ECM interactions. The hydrogel poly[2-(methacryloyloxy)ethyl dimethyl-(3-sulfopropyl)ammonium hydroxide] (PMEDSAH), and polymer coating aminopropylmethacrylamide (APMAAm) have demonstrated iPSC expansion with maintained pluripotency, but only APMAAm is capable of being sterilized using common techniques but required growth with fetal bovine serum to enable stem cell expansion [178,181]. Overall, no ideal 2D ECM has been discovered; moreover, 2D cellular expansion is limited by cell growth surface area and would likely not be capable of expanding iPSCs up to the required scale of 10^7^ to 10^9^ cells. However, discovery that stem cells could be grown and expanded as spheroid clumps in 3D suspension culture has enabled significant scale-up. Previously, stem cells required micro carriers to enable suspension culture and expansion, which significantly limited cell concentrations and expansion capabilities. However, suspension culture with Rho-associated coiled-coil kinase inhibitors (ROCKi) such as Y-27632 has allowed iPSC re-aggregation and prevention of apoptosis [182,183,184]. Enabling suspension-based iPSC culture and expansion will substantially increase scale-up capabilities, with limitations primarily driven by environmental factors, as discussed below.

#### 5.4.3. Environment

The ideal environment for commercial scale cellular expansion will be automated with ideal oxygen, temperature, pH, and chemical factor conditions. To expand iPSC on a small scale, planar plasma-treated polystyrene tissue culture flasks are a viable, economical option [169]. However, for larger-scale expansion, planar growth is expensive, requires highly trained operators, and necessitates large GMP facilities [169,174]. With the advent of 3D stem cell expansion, bioreactors present a favorable option to allow automation, standardization, and reproducibility [169]. More importantly, they allow increased culture surface per volume by removing all the gas layers typically required for oxygenation when using cell stacks. They do this by conducting continuous nutrient replenishment, biochemical (pH, temperature, etc.) control, and waste disposal with recirculated culture medium. This also eliminates open processes, which require large clean rooms as necessitated in flask-based cultures [169]. Complete automation, as with the Lonza Cocoon platform, may provide adequate expansion, with personnel savings, standardization, and cost efficacy provided through an economy of scale [169,174]. It appears that with large volume demand, micro carrier technology is most economical with costs approximately USD 700/dose of 10^9^ cells, a value that could be further improved with greater growth concentrations and lower growth media costs [174]. One barrier that remains is recovery of expanded iPSC cells. Following expansion, cells are washed and centrifuged for recovery. Typical centrifuges technologies shear cells and are not suitable for iPSC retrieval, requiring new technology such as closed continuous fluidized bed centrifuges to be optimized for retrieval [174]. With the advent of 3D iPSC expansion, bioreactors will almost definitely be used to produce the large volume cells for transplant. Once therapeutic success has been achieved with iPSC-based ISCT, bioreactor-based proof-of-concept first-in-human trials will occur soon after.

## 6. Conclusions

Subcutaneous insulin treatment remains the mainstay of DM1 treatment. It enables sufficient carbohydrate metabolism for patients to survive, but remains far from ideal. DM1 patients suffer from hypoglycemia, hyperglycemia, and associated complications that limit their quantity and quality of life. These complications persist despite novel technologies for glycemic monitoring and control. ICT has long provided hope for a cell-based cure. It continues to demonstrate advances with improved glycemic stability, less hypoglycemia, and improved DM-related complications. However, islet engraftment and long-term insulin independence remains approximately 50% and patients must be exposed to potential risks associated with lifelong immunosuppressant therapy. Deceased donor islet sources and funded access also remain limited.

Stem cells derived from hESCs or iPSCs, can be differentiated into mature insulin producing islet cell clusters capable of fully reversing diabetes in mice and rats. However, as ICT transforms from a deceased donor to hESC- or iPSC-based source, several questions will need resolution. Autologous versus allogeneic iPSC sources face unique challenges. Immunosuppression remains a barrier for allogeneic iPSCs, whereas allogeneic sources facilitate scale up with creation of HLA-matched iPSC-derived islet cell cluster banks that can be standardized. Results from ViaCyte’s clinical trials demonstrating successful allogeneic islet maturation and resultant detectable C-peptide levels that correlate with persistence of polyhormonal islet cells within subcutaneous devices provides enthusiasm that immunoprotected allogeneic ISCT is within reach. On the other hand, autologous iPSCs enable immunosuppression-free ISCT, but may be difficult to scale up with such personalized medicine. Further evidence is also required to determine the safety and efficacy of other transplantation sites for ISCT in comparison with the intraportal site, particularly in terms of its potential for off-target growth. Generation of iPSC-derived islet cell clusters with inducible kill switches is also important to consider in this discussion. With these answers, clinicians will require collaboration with multiple parties in the government and industry to standardize GMP protocols, enable consistent international regulations, and create economies of scale. This will likely be enabled with bioreactors utilizing 3D culture expansion of iPSCs, regardless of allogeneic or autologous sources. Regionalization with economies of scale will then enable economic generation of curative therapy for DM. It is certainly an exciting time as we border a new frontier in diabetes care, transitioning from treatment to cure.

## Figures and Tables

**Figure 1 cells-10-00278-f001:**
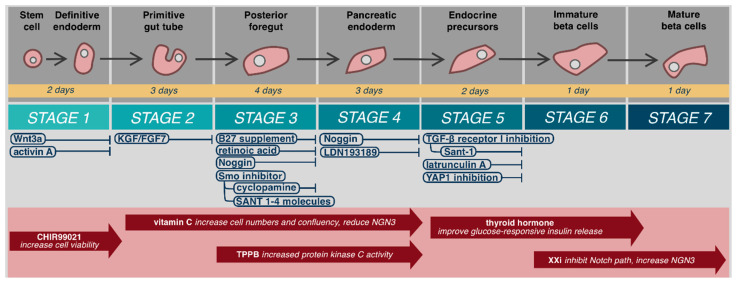
Embryological differentiation and maturation of islet cells.

**Figure 2 cells-10-00278-f002:**
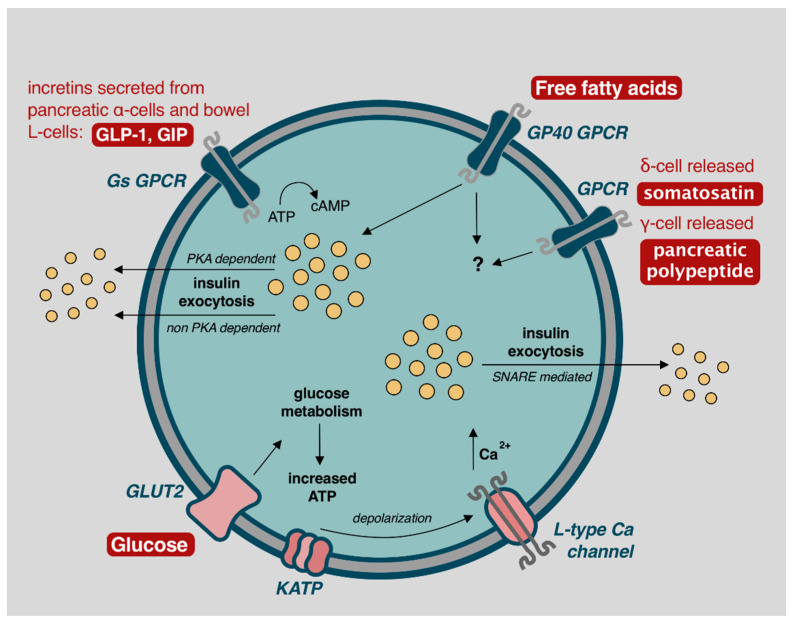
Mechanisms of β-cell insulin release and glycemic control. Image adapted from Komatsu et al. (2013) with permission for reuse [56].

**Figure 3 cells-10-00278-f003:**
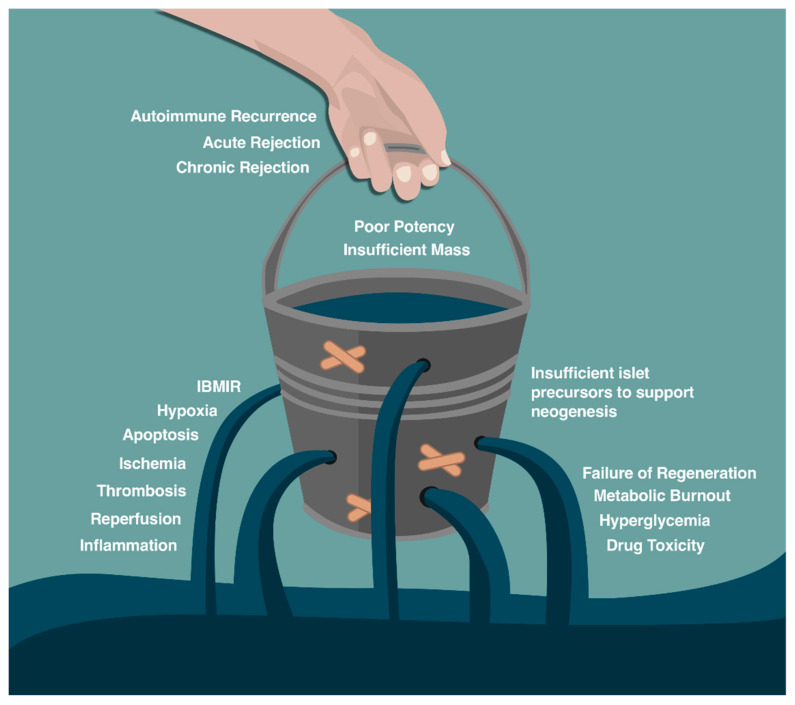
Limiting factors for islet cell engraftment after islet cell transplant. Adapted from Shapiro et al. (2011) with permission for reuse [11].

**Figure 4 cells-10-00278-f004:**
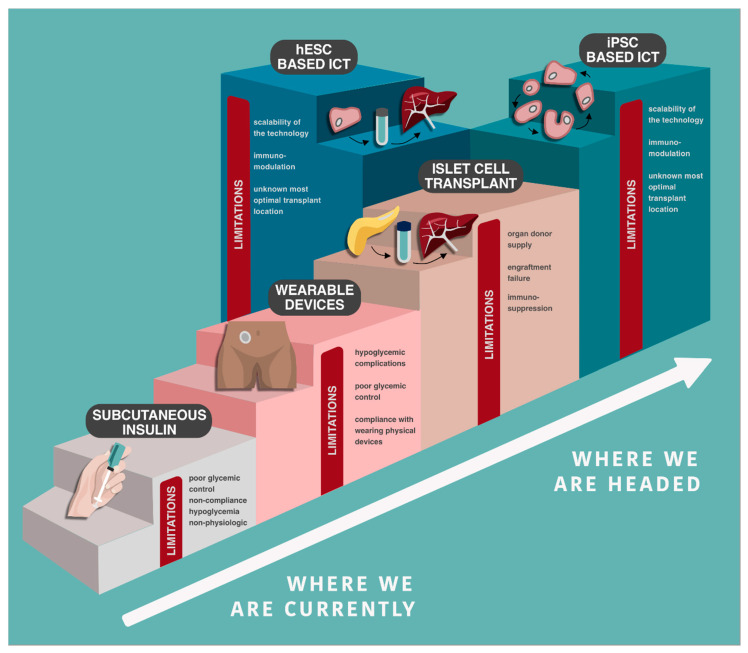
Comparison and advancement of subcutaneous insulin delivery, islet cell transplant, and novel inducible pluripotent stem cell-based islet cell transplant for cure of diabetes.

**Figure 5 cells-10-00278-f005:**
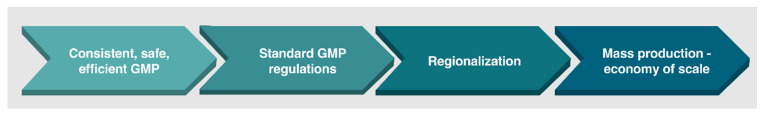
Steps to achieve widespread use of inducible pluripotent stem cell-based islet cell transplant.

**Table 1 cells-10-00278-t001:** Benefits and drawbacks of novel subcutaneous insulin monitoring and delivery devices.

Technology	Benefits	Drawbacks
Continuous glucose monitoring	➢ Immediate glycemic feedback. ➢ Improved dynamic glycemic understanding (real time glycemic targets, time spent in euglycemia, hypoglycemia, or hyperglycemia) [5,12,13] ➢ Hyper/hypo glycemic alarms.	➢ Device discomfort [25]. ➢ Disrupted sleep (alarms) [27].
Continuous subcutaneous insulin infusion (i.e., insulin pump)	➢ Improved glycemic control compared to standard subcutaneous insulin [14,15].	➢ Modest HbA1c improvements (0.3–0.7%) ➢ Mechanical Failure (64% of devices over 7 days). ➢ Device discomfort [25].
Closed loop, wearable insulin delivery device (i.e., artificial pancreas)	➢ Improved glycemic control compared to CSII or CGM [15]. ➢ Improved nighttime hyper- and hypoglycemic control [15].	➢ Poorly accessible. ➢ Device discomfort [25]. ➢ Only 16.4% of the time spent in normoglycemia [15].

CGM: continuous glucose monitor, CSII: continuous subcutaneous insulin infusion.

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
