# Peer review of "Inducible Pluripotent Stem Cells as a Potential Cure for Diabetes"

_cells, 2021, doi:10.3390/cells10020278_

Round 1

Reviewer 1 Report

The paper by verhoeff and colleagues is an in-depth review of the field of transplantation of insulin-producing cells for the treatment of diabetes. I find it truly admirable how the authors have managed to approach the subject with this completeness, from insulin delivery to islet transplantation, from the differentiation of pluripotent stem cells to the limits of their application to a large audience of people with diabetes.
The bibliography is extensive and accurate.
I would like to advise the authors to indicate Viacyte's NCT trial number on page 11. In the same paragraph it would perhaps be useful to add something more on the encapsulation strategy, in particular the Anderson studies (PMID: 32231313, PMID: 30873298, PMID: 26808346) are worthy of mention, having also laid the foundations for the future clinical application of Sigilon Therapeutics. Figure 1 is difficult to read because of the very small font used. 

Reviewer 2 Report

  1. It would be helpful if the authors could include whether or not any adverse effects have been observed with any of the immunosuppressive drugs which have been used in ICT.
  2. Page 5 line 229: It is wrong to state that "in response, ATP-dependent K+ channels (KATP) channels open". Both K+ and Ca++ channels cannot open simultaneously during depolarization. With increase in ATP the K+ channels are inhibited while the Ca++ channels open for Ca++ influx into the beta cell to trigger exocytosis.
  3. Page 14 line 578: The sentence beginning on this line should be modified as it gives the impression that all diabetic patients need to be treated with ICT. ICT will always be limited to type 1 diabetic patients and the approximately 25% of individuals afflicted with type 2 diabetes being treated with insulin.
  4. Conclusion section: The sentence beginning on line 717 should be checked for accuracy - did they authors mean to say "independence" rather than dependence?
  5. The manuscript should be checked for grammatical errors and "typos". For instance the sentence on page 5 line 205 needs correction.
  6. All the references should be checked for accuracy and completion as reference #6 is shown as "invalid citation". Also, reference #14 appears to be incomplete with no authors.
